# Oxygen for the Newborn: Friend or Foe?

**DOI:** 10.3390/children10030579

**Published:** 2023-03-17

**Authors:** Serafina Perrone, Sara Manti, Chiara Petrolini, Valentina Giovanna Dell’Orto, Giovanni Boscarino, Chiara Ceccotti, Mattia Bertini, Giuseppe Buonocore, Susanna Maria Roberta Esposito, Eloisa Gitto

**Affiliations:** 1Neonatology Unit, Pietro Barilla Children’s Hospital, Department of Medicine and Surgery, University of Parma, 43126 Parma, Italy; 2Pediatric Unit, Department of Human Pathology “Gaetano Barresi”, University of Messina, 98122 Messina, Italy; 3Pediatric Clinic, Pietro Barilla Children’s Hospital, University of Parma, Via Gramsci 14, 43126 Parma, Italy; 4Department of Molecular and Developmental Medicine, University of Siena, 53100 Siena, Italy; 5Neonatal Intensive Care Unit, Department of Human Pathology “Gaetano Barresi”, University of Messina, 98122 Messina, Italy

**Keywords:** preterm, reactive oxygen species, reactive nitrogen species, free radicals, newborn infants, prematurity

## Abstract

Oxygen supplementation is widely used in neonatal care, however, it can also cause toxic effects if not used properly. Therefore, it appears crucial to find a balance in oxygen administration to avoid damage as a consequence of its insufficient or excessive use. Oxygen toxicity is mainly due to the production of oxygen radicals, molecules normally produced in humans and involved in a myriad of physiological reactions. In the neonatal period, an imbalance between oxidants and antioxidant defenses, the so-called oxidative stress, might occur, causing severe pathological consequences. In this review, we focus on the mechanisms of the production of oxygen radicals and their physiological functions in determining a set of diseases grouped together as “free radical diseases in the neonate”. In addition, we describe the evolution of the oxygenation target recommendations during neonatal resuscitation and post-stabilization phases with the aim to define the best oxygen administration according to the newest evidence.

## 1. Introduction

Oxygen is essential for aerobic life but, as a drug, has both biological benefits and toxic effects [1]. In the neonatal period, oxygen supplementation can be essential to treat acute and chronic conditions associated with hypoxemia. However, oxygen overexposure may lead to important consequences [1].

Oxygen toxicity happens in case of oxidative and nitrosative stress, a condition where the levels of reactive oxygen/nitrogen species (ROS/RNS) overwhelm the capacity of antioxidant defenses leading to potential damage through reaction with lipids, proteins, DNA, amino acids and several other molecules [2]. This condition can be caused by either increased ROS/RNS formation, decreased activity of antioxidants, or both [2,3].

In order to describe the mechanisms of the production of oxygen radicals, their physiological functions, and their pathological role in determining a set of pathologies grouped together as “free radical diseases in the neonate”, a narrative review was conducted.

In addition, we reported the evolution of the oxygenation target recommendations during neonatal resuscitation and post-stabilization phases with the aim to define the best oxygen administration according to the newest evidence.

## 2. Methods

### 2.1. Research Strategy

Two reviewers (SP, SM) independently conducted searches of electronic medical literature databases, such as PubMed and Science Direct databases. On these websites, we searched for articles from 1 January 2012 to December 2022, using the following key terms: “oxygen”, “oxidants”, “reactive species”, “reactive oxygen species”, “ reactive nitrogen species”, “antioxidants”, “non-enzymatic antioxidants”, “retinopathy of prematurity”, “periventricular leukomalacia”, “bronchopulmonary dysplasia”, “necrotizing enterocolitis”, “patent ductus arteriosus”, “newborn”.

### 2.2. Study Selection

Articles were included in the review according to the following inclusion criteria: English language, publication in peer reviewed journals, and year of publication being at least 2012. Editorials, commentaries, case reports, and case series were excluded from the analysis. Articles were excluded by title, abstract, or full text for irrelevance to the investigated issue. Lastly, to identify further studies that met the inclusion criteria, the references of the selected articles were also reviewed. The included studies in this review focusing on oxygen toxicity and free radical diseases are summarized in Figure 1.

## 3. Reactive Oxygen and Nitrogen Species

Reactive species include two classes of chemically-reactive molecules containing oxygen (reactive oxygen species, ROS) and nitrogen (reactive nitrogen species, RNS) which are collectively termed reactive oxygen and nitrogen species (RONS). RONS comprise both free radical and non-free radical oxygenated molecules [4].

The majority of RONS is composed of free radicals, that is, any chemical species capable of independent existence that contains one or more unpaired electrons [5]. The presence of unpaired electrons makes free radicals unstable molecules, which can cause chain reactions involving many steps, each one forming a free radical that triggers the subsequent reaction [6].

Radicals can be oxidized or reduced. The defining feature of radicals is that they react rapidly with each other. So they go away quickly. As the products of radical molecule reactions are again radicals, sequential reactions are a natural fit.

The fact that a given radical can be observed does not mean that it is an intermediate on the pathway from precursors to products. Indeed, chain reactions tend to produce more persistent radicals. In turn, these are present in higher concentrations and, therefore, are easier to detect than the transient radicals involved in fast propagation steps.

Radical chain reactions are innate (or natural) cycles that operate without a catalyst. If the propagation steps in a given cycle are rapid enough, then reactions will occur provided that there is a suitable mode of initiation and absent inhibition.

Unlike many catalytic cycles, innate radical chain cycles do not start spontaneously; they require initiation, which is the generation of radicals from nonradicals. Furthermore, initiation must generate a radical in the chain, not just a radical [7].

Termination of this chain reaction can occur when two free radical species react with each other to form a stable and non-radical compound [6].

Reactive oxygen species are small molecules formed as a natural product of the normal metabolism of oxygen. Free radical ROS include superoxide anion (O_2_^•−^), hydroxyl radical (OH^•^), peroxyl radical (ROO^•^), lipid peroxyl (LOO^▪^) and alkoxyl radical (LO^▪^) [8]. Non-radical ROS, which are either oxidizing agents or easily converted into radical species, are hydrogen peroxide (H_2_O_2_), lipid peroxide (LOOH), hypochlorous acid (HOCl), ozone (O_3_) and singlet oxygen. Singlet oxygen can exist in two states, the delta state (^1^Δ_g_) and sigma state (^1^Σ_g_+) [8]. The first one is a non-radical while the second one is a free radical because it contains two unpaired electrons; however, the sigma state is highly unstable in nature therefore has no biological significance [8].

Hydroxyl radical is extremely reactive, so that it reacts close to its site of formation; otherwise superoxide anion and hydrogen peroxide are less reactive [8].

Reactive nitrogen species are nitrogen-containing molecules generated through the interaction of free radical nitric oxide (NO) with reactive oxygen species such as superoxide and hydrogen peroxide [9]. They include nitric oxide (^•^NO), peroxynitrite (ONOO^•^), nitrogen dioxide radical (NO_2_^▪^) and peroxynitrous acid (HNO_3_) [9].

Primary RONS, such as superoxide anion, hydrogen peroxide or nitric oxide, are less toxic as they react reversibly with biomolecules, and their effects are mitigated by enzymatic antioxidants [4]. Damage is essentially due to secondary RONS, such as hydroxyl radical, peroxynitrite and hypochlorous acid, as there is no specific enzymatic system dedicated to control their levels [4].

The main source of ROS is represented by mitochondria, intracellular organelles that play a leading role in adenosine triphosphate (ATP) production through oxidative phosphorylation [10]. Oxidative phosphorylation is a metabolic pathway constituted by two parts, the electron transport chain (ETC) and the chemiosmosis. The ETC is a series of four protein complexes (complex I-II-III-IV, coenzyme Q and cytochrome C) bound to the inner mitochondrial membrane, that couple redox reactions, creating an electrochemical gradient that leads to the generation of ATP. The electrons come from breaking down organic molecules and energy is released.

The energy released forms a proton gradient, which is used in chemiosmosis to make a large amount of ATP by the protein ATP-synthase.

Under physiological conditions, about 2% of electrons do not follow the normal transfer order but leak out of ETC causing a partial reduction of oxygen and producing three intermediate products as superoxide anion, hydrogen peroxide and hydroxyl radical [11].

Both complex I (NADH dehydrogenase) and complex II (succinate dehydrogenase) of the oxidative phosphorylation process have been demonstrated to generate free radicals [12]. Complex III is also a major site for the production of ROS by the ETC via the direct reduction of molecular oxygen to form superoxide. Reductants enter complex III via the endogenous quinol pool from complex I or via exogenous donors to complex III.

A significant role in oxidative stress is also played by Fenton and Haber–Weiss reaction which occurs especially with the presence of an iron overload [13]. Iron is important for proper growth and neurological development, but can be toxic when unbound [14], (Figure 2).

In moderate quantities and when bound to transport or storage proteins, such as transferrin and ferritin, iron is an essential element for cell aerobic metabolism and growth.

During clinical conditions of hypoxia, ischemia, acidosis, iron escapes from these proteins and redox active non protein bound iron, Fe^2+^, in the presence of oxygen, enters the reaction of Fenton and produces hydroxyl radical, the most powerful oxidizing agent of a biological system.

Free iron is a highly reactive element that can act both as a strong biologic oxidant and a reducing agent thanks to its ability to switch back and forth between ferrous and ferric oxidation states by varying the ligands with which it is associated [15]. Iron toxicity is mitigated by its tissue storage in protein such as ferritin and hemosiderin and its sequestration in transport proteins such as transferrin to which the element is bound at high-affinity binding sites [15]. In case of iron excess, the binding sites of plasma transferrin become completely saturated and mobile pool of non-transferrin bound iron can react with reduced intermediate products of oxygen generating harmful oxidants [15].

In Fenton reaction the ferrous iron (Fe^2+^) reacts with hydrogen peroxide producing ferric iron (Fe^3+^) and hydroxyl radical ($Fe^{2+}+H_{2}O_{2\;}\to Fe^{3+}+OH\cdot +OH^{-}$) [16]. Ferric ion is then reduced back to ferrous ion thanks to the interaction with superoxide anion radical ($Fe^{3+}+\cdot O_{2}^{-}\;\to \;Fe^{2+}+O_{2}$) [17]. In the Haber–Weiss net reaction superoxide anion radical reacts with hydrogen peroxide producing hydroxyl radical, hydroxyl anion and molecular oxygen $\left(\cdot O_{2}^{-}+H_{2}O_{2\;}\to O_{2}+OH\cdot +OH^{-}\right)$ [13].

Plasma membrane is another important source of ROS due to the presence of the membrane-bound enzyme NADPH oxidases (NOX) belonging to the NOX family [17]. NOX family enzymes share the capacity to transport electrons across the plasma membrane generating superoxide and other downstream ROS, however, they differ in organ-specific expression, subcellular localization, regulation of expression, and activity and type of ROS released [17].

There are seven NOXs: NOX1–5, dual oxidase 1 (Duox-1) and Duox-2 [18]. All the NOXs except NOX5 require p22phox, a membrane-bound subunit, to be activated and generate ROS. NOX5 activation does not require cytosolic regulatory subunits because it is activated by intracellular calcium. NOX1, NOX2, and NOX5 can each be activated and individually reduce oxygen to O_2_^•−^, which is eventually converted to H_2_O_2_ by superoxide dismutase.

The main production of ROS by NOX occurs in phagocytes activated by exposure to microbes or mediators of inflammation which generate a high amount of superoxide anion and hydrogen peroxide [18]. Microorganisms killing is not only a direct consequence of toxic effects of reactive oxygen species produced by NOX family of phagocytes. In fact these enzymes are also able, using their ROS production, to finely regulate the immune response as they can inactivate microbial virulence factors through a hydrogen peroxide myeloperoxidase system, ensure normal functioning of proton channel and regulate peroxisomal pH to guarantee optimal activity of neutral proteases in neutrophils [19,20,21]. At last, oxidative stress during inflammation is enhanced by the increased release of NO which can be a source of free radical as it contains an unpaired electron in its outer orbital [22].

Expression of NOXs can be modulated by hypoxia, proinflammatory cytokines such as TNFα, TGFβ, and IFNγ, or by transcriptional and epigenetic mechanisms [23]. Additionally, a number of transcriptional factors are identified to regulate gene expression of Noxs, including hypoxia-inducible factor 1 subunit alpha (HIF-1α), nuclear factor kappa B (NF-κB), nuclear factor-erythroid factor 2-related factor 2 (Nrf2) and signal transducer and activator of transcription [24].

ROS production increases in pathologic condition such as ischemia, ischemia/reperfusion and hyperoxia, potentially leading to cellular damage. Ischemia enhances ROS production inducing a fast change in the redox state of mitochondrial complexes, resulting in electron leak to oxygen [25]. In case of reversible ischemia, a large generation of ROS occurs quickly with the reestablishment of blood supply [26]. Hyperoxia may also increase ROS production: increasing partial pressure of O_2_ itself is not sufficient to oxidize biomolecules, and hyperoxia-induced cellular damage is likely due to increased partial reduction of molecular oxygen to superoxide radical in metabolically dependent reactions [27].

### Functions of ROS

ROS, considered in the past as a harmful and unwanted product of cellular metabolism, are actually known to be essential in regulation of physiological cellular functions as they are involved in several processes such as gene expression, signal transduction pathways, cell fate decisions, energy metabolism and protein import or folding [28,29]. ROS participate in the differentiation and self-renewal of stem cells: a low level of ROS selects more potent hematopoietic stem cells while their accumulation controls proliferation and self-renewal in neuronal stem cells [28,29]. ROS also contribute to both innate and adaptive immune responses. In innate response they coordinate leukocytes diapedesis at the beginning of a phlogistic process, allowing microorganisms elimination through phagocytosis, and are also involved in controling the resolution of inflammation [30]. In adaptive response, ROS mediate the activation of T cells and has been suggested to have an immunosuppressive role [31]. In the kidney, hydrogen peroxide acts as a regulator of the vessel wall tension ensuring the proper functioning of kidney vasculature [32]. A final example of the numerous functions played by ROS concerns platelet generation, as they contribute to the correct maturation of megakaryocytes [33].

## 4. Antioxidants

Antioxidants neutralize free radicals inhibiting the oxidation of any molecule, they can be divided into enzymatic and non-enzymatic antioxidants.

### 4.1. Enzymatic Antioxidants

The main enzymatic antioxidants in newborns are superoxide dismutase (SOD), catalase (CAT), and glutathione peroxidase (GPx) [34].

SOD detoxifies the superoxide anion by converting two molecules of superoxide radical into molecular oxygen and hydrogen peroxide $\left(2O_{2}^{-}\;\to O_{2}+H_{2}O_{2}\right)$. Hydrogen peroxide is a harmful product of many normal metabolic processes, which is quickly converted into other, less dangerous, substances in order to prevent damage. At this point, catalase catalyse the decomposition of hydrogen peroxide into water and molecular oxygen $\left(2H_{2}O_{2}\to 2H_{2}O+O_{2}\right)$ [34].

GPx reduces hydrogen peroxide to glutathione disulphide and water. Unlike SOD and CAT, GPx requires several secondary enzymes (glutathione reductase and glucose-6-phosphate dehydrogenase) and cofactors (monomeric glutathione, NADPH, glucose 6-phosphate) to function. The main reaction that glutathione peroxidase catalyzes is $2GSH+H_{2}O_{2}\to GS-SG+2H_{2}O$ [35]. The reaction begins with the oxidation of the selenol of a selenocystein residue (RSeH) by hydrogen peroxide and the formation with water and a selenenic acid group (RSeOH), the latter is then converted back to selenol by a two-step process that begins with reaction with monomeric glutathione (GSH) to form GS-SeR and water. A second GSH molecule reduces the GS-SeR intermediate back to selenol, releasing glutathione disulphide (GS-SG) by-product. Glutathion reductase, with NADPH as a cofactor, then reduces the oxidized glutathione to complete the cycle [35].

### 4.2. Non-Enzymatic Antioxidants

Non-enzymatic antioxidants, mainly constituted by vitamins, are classified into water-soluble and lipid-soluble. Vitamin C, the most studied among water-soluble vitamin, prevents teratogenic effects on maternal diabetes similarly to vitamin E [36].

Vitamin C is a powerful antioxidant preventing lipid peroxidation in plasma exposed to oxidative stress. Preterm infants are born with high plasma vitamin C concentrations, which drop soon after birth. In the presence of redox-active iron, vitamin C can act as a pro-oxidant in vitro and contribute to the formation of hydroxyl radicals, which in turn may cause lipid, DNA, or protein oxidation [37]. Thus, Silvers et al. suggested that high plasma concentrations of Vitamin C in preterm infants are harmful because of the presence of free iron in preterm infants [38].

Berger TM et al. [39] demonstrated that endogenous and exogenous vitamin C delayed the onset of iron-induced lipid peroxidation in a dose-dependent manner. The Authors reported that in iron-overloaded plasma, vitamin C acts as an antioxidant toward lipids. The findings did not support the hypothesis that the combination of high plasma concentrations of vitamin C and free iron causes oxidative damage to lipids and proteins in vivo [39].

Some of the effects of maternal smoking on lung development can be prevented by supplemental vitamin C [40].

Uric acid can be considered a scavenger of many ROS, anyway, it can also have pro-inflammatory effects, playing a role in the pathogenesis of diseases such as diabetes, if present in excessive amounts [37]. Ceruloplasmin plays a role of extracellular antioxidant by catalyzing the oxidation reaction of iron in a similar way to the ferroxidase enzyme [37].

Among the lipid-soluble, vitamin E is the antioxidant that is present in the highest amount in the human body and plays an important role as a lipid peroxyl scavenger [37].

Chemically, α -tocopherol is the most active form of vitamin E due to the substitution of the methyl groups on the chromanol ring which makes the hydrogen of the C-6 hydroxy group particularly active, facilitating the transfer of hydrogen to a peroxyl radical. α-tocopherol therefore has a higher rate of transfer of hydrogen atoms to a fatty acid peroxyl radical than any other fat-soluble antioxidant [41].

Lipid peroxidation follows the following process: “Initiation”, “Propagation”, and “Termination”.

Briefly, ROS, such as (HO^•^) act as pro-oxidants, and unsaturated fatty acid (LH) forms a lipid radical (L^•^). The reaction rate depends on the presence of an initiator, and on the chemical properties of the initiator. In the “Propagation” step, (L^•^) reacts with oxygen to produce a lipid peroxyl radical (LOO^•^). This reaction rate is fast because (L^•^) is very reactive. Then, (LOO^•^) slowly reacts with (LH) to form lipid hydroperoxide (LOOH). Once lipid peroxidation progresses, the chain reaction does not stop until the final product is formed in the “Termination” step. In the process of forming such lipid peroxides, the presence of α-tocopherol in the reaction delivers hydrogen ions to (LOO^•^), which itself becomes a radical, causing further lipid peroxidation [42,43].

This phenomenon called “tocopherol-mediated peroxidation” exemplifies the paradoxical role of α-tocopherol in lipid autoxidation.

There is also a second possibility by which α-tocopherol can exert a pro-oxidant activity. This comes from its ability to reduce transition metal ions to their lower oxidation states, e.g., Fe^2+^ and Cu^+^. The reduced metal ions have a higher rate of reaction with hydroperoxide and hydrogen peroxide than the oxidized forms. The reactions can result in the cleavage of the hydroperoxides and lead to the generation of reactive alkoxyl radicals [44].

The antioxidant activity of α-tocopherol has led to the expectation that dietary supplementation with vitamin E may prove to be a beneficial measure against lipid oxidation. While this hypothesis appears compelling and straightforward, data from clinical trials of vitamin E supplementation in preterm infants have provided little supporting evidence. Most of the investigations on whether vitamin E reduces the severity or inhibits the development of free radical-mediated diseases in preterm infants resulted in a misfire, with a few exceptions for intraventricular hemorrhage prevention. However, almost all these studies were performed from the 1980s to the early 1990s, Therefore, these results are difficult to apply in modern neonatal care.

## 5. Newborn Susceptibility to Oxidants

Newborns, especially preterm ones, are more susceptible to oxidative stress than individuals later in life for several reasons.

The first one is represented by the hypoxic–hyperoxic challenge during the transition to intrauterine and extrauterine life. Fetal growth requires a relatively hypoxic environment, the uterus, with 20–25 mmHg of oxygen tension (PO_2_). After delivery, the newborn is exposed to an extrauterine normoxic environment of approximately 100 mmHg PO_2_ [45]. If supplementary oxygen resuscitation is required during the fetal-to-neonatal transition, the fetus may even be exposed to a hyperoxic environment. The abrupt increase in oxygen availability, after a period of lack of oxygen, causes the physiologic production of oxygen species and may be harmful through the production of ROS in a mechanism called “hypoxia-reperfusion injury” [46]. During ischemia, the degradation of ATP results in the production of hypoxanthine and the lack of ATP production causes ATP-dependent ionic pumps, like Ca_2+_pumps, to fail with loss of transmembrane ionic gradients with subsequent release of calcium from the mitochondria into the cytoplasm and extracellular spaces. This induces the activation of mitochondrial calcium-dependent cytosolic proteases including calpain, which convert the cellular enzyme xanthine dehydrogenase to xanthine oxidase. Once the ischemic tissue is reperfused, an influx of molecular oxygen catalyzes xanthine oxidase to degrade hypoxanthine to uric acid thereby liberating the highly reactive superoxide anion (O_2-_). Superoxide is subsequently converted to hydrogen peroxide (H_2_O_2_) and the hydroxyl radical (OH^•^) [47].

According to these observations, Saugstad et al. showed that hypoxanthine is increased in the blood of hypoxic newborns and increases exponentially in the first minutes of resuscitation. Due to the accumulation of hypoxanthine during hypoxia, the working group suggested limiting the use of oxygen, during resuscitation, to avoid an explosive generation of oxygen radicals [47,48].

Secondly antioxidant capacity is lower in the newborn, especially in the premature infant, therefore premature babies are especially prone to oxidant injury. Plasma levels of enzymatic antioxidants decrease in direct proportion to the reduction of gestational age since antioxidant defenses develop during the last trimester of pregnancy [49]. Enzymatic anti-oxidant values also show a correlation to neonatal development, as their plasma levels are lower in the case of intrauterine growth restriction than in terms adequate for gestational age (AGA) neonates [50,51].

Newborn infants also have a lack in non-enzymatic antioxidants as evidenced by several studies. Bayadas et al. described fewer lipid-soluble antioxidant vitamins (vitamins A and E) in the serum of preterm babies [52].

To a lesser extent, the increased production of free radicals is due to infections and high levels of free irons. Newborns, especially if preterm, can have a distinct immune system and are therefore more prone to infections which result in an increased production of ROS [53]. Plasmatic levels of free iron are higher in newborns than in older children so the production of ROS increases through an enhancement of Fenton and Heber–Weiss reactions [54].

## 6. Free Radical Disease in Newborn

Oxidative stress is now considered the common denominator of several pathologies able to present itself with different manifestations according to the organ mostly affected. Among these, grouped together as “oxygen radical disease of neonatology”, are counted retinopathy of prematurity (ROP), periventricular leukomalacia (PVL), bronchopulmonary dysplasia (BPD), necrotizing enterocolitis (NEC) and patent ductus arteriosus (PDA) [55]. Table 1 shows clinical studies investigation the association between oxidative stress biomarkers and free radical-mediated diseases of newborns.

### 6.1. Retinopathy of Prematurity

Despite improvements in neonatal care, ROP represents one of the main causes of visual impairment and blindness in premature babies. Several perinatal and postnatal factors contribute to its genesis with a higher incidence of decreasing birth weight and gestational age [63].

ROP is a multiphasic pathologic process linked to the incomplete maturation of retinal vascularization in premature newborns. It occurs in two phases, the vascular attenuation phase and the fibrovascular proliferative one in which suppression and production of vascular-endothelial growth factor (VEGF), respectively, plays an important role [56].

The vascular attenuation phase is the first phase of ROP. It starts after preterm birth when supplemental oxygen, required during resuscitation/stabilization in the delivery room, causes cessation of normal retinal vascularization. The enhanced production of ROS, due to supplemental oxygen, determines structural and functional changes in several retinal molecules leading to the arrest of neurovascular development and the obliteration of existing vascular beds [64].

The fibrovascular proliferative phase is the second phase in which hypoxia, due to the arrest of the growth of retinal blood vessels, induces retinal vasoproliferation increasing VEGF production to improve retinal perfusion. Hypoxia itself increases ROS slowing upstream events in the electron transport chain, raising the concentration of oxygen donors and promoting oxidative stress [65].

The retina is highly sensitive to lipid peroxidation as it is composed of lipids with high levels of polyunsaturated fatty acids (PUFAs), such as docosahexaenoic acid (DHA), cis-arachidonic acid, and choline phosphoglyceride [66].

Among a panel of biomarkers currently investigated, MDA, GSH/GSSG, 8-Hdhg, and PUFAs appear to be the most promising for the diagnosis and treatment monitoring of ROP [58,67]. All these biomarkers are involved in the cascade of events leading to oxidative stress damage. Thus, their potential role in the prevention of retinal damage and irreversible vision disorders is justified.

### 6.2. Periventricular Leukomalacia

PVL is the main substrate for cerebral palsy and is a form of cerebral white matter damaged characterized by focal necrosis of white matter near the lateral ventricles, with subsequent cystic formation, and more diffuse injury of deep cerebral white matter. The classic neuropathology of PVL has given rise to several hypotheses about the pathogenesis, largely relating to ischemia-reperfusion damage in the premature infant [68].

Oligodendroglia (OL) precursor cells, the main target in PVL, are exquisitely vulnerable to attack by free radicals [59]. This vulnerability is maturation-dependent, decreasing in OL mature cells, and appears to relate to a developmental window characterized by a combination of deficient antioxidant defenses and active acquisition of iron during OL differentiation [68]. The high concentration of unsaturated fatty acids in the neonatal brain and the high metabolic rate, almost exclusively supported by oxidative metabolism of the developing neural tissues, predispose to the propagation of oxidative stress [59].

Coviello et al. [60] demonstrated that measurement of plasma-isoprostanes in the first days of life in very low birth weight infants may help discriminate patients showing abnormal white matter injury score at the term of gestational age, thus showing the potential to early identify newborns at risk for brain injury.

Plasma-isoprostanes may also represent a useful biomarker of brain vulnerability in high-risk infants [69].

### 6.3. Bronchopulmonary Dysplasia

BPD is the major cause of chronic lung disease and morbidity in preterm infants. As opposed to “old BPD”, expression of lung injury caused by high airway pressures and oxygen toxicity, the “new BPD” is the result of complex interactions between altered alveolar and vascular development, injury by ante-/post-natal pathogenic factors and lung reparative processes [62,70]. One of the main causes of BPD is oxidative stress continuously sustained by the lung inflammatory process due to hyperoxia exposure. Inadequate nutrition and type of ventilation contribute to increased oxidative stress which may trigger changes leading to permanent lung damage [1,71]. Hyperoxia-induced acute oxidative stress is mediated by the increased number and over-activation of NOX1 during hyperoxia with subsequent excessive ROS production and destruction of the alveolar capillary barrier [57].

Oxidative stress can lead to BPD in various ways. Fibrotic process described in BPD is due to the balance between matrix metalloproteins (MMPs) and their inhibitors. Oxidative stress increases both MMPs and their inhibitors, so it may lead to lung damage by increasing collagenase activity causing disruption of the extracellular matrix [61].

Oxidative stress also interferes with the production of surfactant. As a result of inflammation and edema, transudate plasma proteins and inflammatory cells impair extracellular surfactant. Hyperoxia reduces also surfactant phospholipid production [72].

Furthermore, alveolar epithelial type-II cells (AEC II) are the most important stem cells in lung tissue playing a crucial role in the regulation of lung tissue growth, maturation, wound-repair process, and lung fluid homeostasis. The proliferation and differentiation of AECIIs are the major factors that regulate the post-injury repair of alveolar epithelial structure and functions. Additionally during lung injury, as a stem cell, AECII can be converted to AECI, and can secrete a variety of bioactive substances to restore alveolar capillary barrier integrity and maintain alveolar function. Oxidative stress due to hyperoxia exposure interferes with all these important functions of AECII inducing excessive apoptosis of AECII and inhibiting their proliferation, promoting further lung damage [73].

The involvement of harmful effects of oxygen supplementation and oxidative stress has been demonstrated in experimental and clinical studies in the neonatal population [74].

### 6.4. Necrotizing Enterocolitis

NEC is a disorder of premature newborns that results in inflammation and bacterial invasion of the bowel wall [75]. Many etiologic factors including immaturity, hypoxia/ischemia, hyperosmolar feedings, bacterial colonization, and oxidative stress have been identified [75]. Some authors recently proposed that the underlying initial condition is the reduced ability of the neonatal gut epithelial cells to reduce oxidative stress so that, after bowel exposition to enteral feeding, the increased oxidative stress tips the epithelial gut cells toward apoptosis, inflammation, bacterial activation, and eventual necrosis [76].

Confirming the role played by oxidative stress in determining NEC, levels of non-protein bound iron and total hydroperoxides, measured in cord blood, showed a clear correlation between intrauterine stress oxidative events and the risk of developing NEC [77].

### 6.5. Patent Ductus Arteriosus

PDA consists of a persistence of the fetal communication between the descending aorta and the left pulmonary artery. After birth ductus arteriosus usually closes in the first days of life and closure is due both to a decrease of circulating prostaglandin E2, a vasodilator synthetized by the placenta during fetal life, and an increase of circulating oxygen after breathing [78]. Ductus arteriosus (DA) is therefore affected by changes in pO2 as intrauterine hypoxia maintains the patency of the ductus and hyperoxia at birth facilitates its closure.

PDA is more frequently seen in preterm infants, particularly those born at <30 weeks’ gestation [78], probably due to the different role played by isoprostanes (IsoPs), a biomarker of oxidative stress, in preterm or term newborns [79]. Oxygen exposure increases IsoPs levels in newborn mouse lungs causing constriction of the isolated DA by activating the thromboxane (TxA2) receptor, however, IsoPs induce vasodilation of preterm isolated DA following the interaction with prostaglandin E2 receptor 4 (EP4). IsoPs can therefore mediate constrictive or dilatory effects on the DA according to the relative concentration of TxA2 and EP4 receptors which modify their levels in consideration of gestational age. In fact, as gestational age increases, EP4 receptors are reduced and TxA2 receptors increase. In consideration of these studies, it can be hypothesized that oxidative stress associated with delivery could favor DA closure in term infants by activating the TxA2 receptors or DA dilation and PDA in preterm infants by activating EP4 receptors [80].

In addition, hemodynamically significant PDA may cause hypoperfusion of organs which can contribute to the genesis of other free radical diseases [81,82]. In line with these findings, Longini et al. [83] reported increased urinary IsoPs levels in forty-three preterm babies (GA: <33 weeks) with PDA. Moreover, seven days after treatment with ibuprofen, a significant decrease in IsoPs was detected, confirming the role of inflammation in the PDA and the benefit of ibuprofen treatment in reducing the risk of OS related to free-radical (FR) generation.

## 7. Discussion

Oxygen has been used uncritically during neonatal resuscitation and care until a few years ago. For example, in 1992, the American Heart Association’s guidelines for newborn resuscitation stated that “100% oxygen should be used, it is not toxic, and there were no reasons to be concerned” [84].

Historically, toxicity of oxygen was first hypothesized in 1951 by Dr. Campbell, who correlated oxygen supplementation to a new form of blindness in premature infants successively entitled ROP [85].

Early clinical studies during the mid-twentieth century also suggested that restricted oxygen use resulted in fewer deaths, decreased respiratory failure, and decreased lung disease compared to liberal oxygen use. According to these data reduced oxygen supplementation was accepted in clinical practice, however, associated with an increased incidence of cerebral palsy. In a multicenter study of 1080 infants weighing < 1800 g, a median duration of supplemental oxygen exposure of <2 days was associated with a 17.4% incidence of cerebral palsy and an 8.7% incidence of ROP whereas a median supplemental oxygen exposure of >10 days was associated with a 5.8% incidence of spastic diplegia but a 21.7% incidence of ROP [86].

The presence of a delicate balance between the toxic and positive effects of oxygen supplementation, secondary to the amount of oxygen administered during neonatal care, made it necessary to carry out further studies in order to define the optimal oxygen values to be administered and target values of blood saturation, especially in preterm infants.

The Supplemental Therapeutic Oxygen for Pretreshold ROP (STOP-ROP) study was one of the leading randomized clinical trials. This study randomized preterm infants with pre-threshold ROP at 35 weeks of gestational age to either 89–94% or 96–99% SpO2. STOP-ROP demonstrated that supplemental oxygen to keep saturations in the 96–99% range did not prevent the progression of pre-threshold disease but increased the risk of adverse pulmonary events including pneumonia, chronic lung disease, the need of diuretics, and hospitalization at 3 months of corrected age [87].

Afterwards, the Benefits of Oxygen Saturation Targeting (BOOST) study randomized infants born before 30 weeks of gestational age, treated for at least two weeks with oxygen supplementation, to either 91–94% or 95–98% SpO2 targets. This trial confirmed that targeting oxygen saturations at higher values did not confer any benefit in terms of the growth and development of premature infants as compared to targeting oxygen saturations to a lower range. A higher target of oxygen saturation was also associated with more necessity of home oxygen [88].

Further studies have been carried out to evaluate both a greater number of patients and the effects associated with a lower oxygen saturation target than in the previous studies.

More data from five clinical studies (SUPPORT, COT, BOOST II UK, BOOST II NZ, BOOST II AU) were processed by the Neonatal Oxygen Prospective Meta-Analysis (NeOProM) in order to define the best target of oxygen saturation in preterm infants randomizing 4965 babies <28 weeks gestational age to a low (85–89%) or a high oxygen saturation target (91–94%) before 24 h of life [89].

The first outcome evaluated by NeOProm was the combination of death and/or major disability/neurodevelopmental impairment such as blindness, deafness, cognitive impairment, or cerebral palsy at 18–24 months of age. Relative risk showed no difference between the low and high saturation target group. Mortality separately showed a significant increase in relative risk in the low saturation target group, otherwise NDI alone, in particular cerebral palsy, showed no difference between the groups. Impairment ROP requiring treatment showed a 25% reduced risk in the low saturation group while severe NEC requiring treatment showed a 25% increased risk in this group. PDA needing treatment, severe intraventricular hemorrhage grade 3–4, deafness and severe visual impairment did not show differences between the groups.

BPD defined physiologically (SUPPORT, BOOST II UK study) showed a non-significant tendency to a small reduction in the low saturation target group. However, when BPD was defined as oxygen requirement at 36 weeks postconceptional age, there was a significant reduction in the low saturation target group [89].

According to NeOProm data, recommendations suggest to keep the saturation target between 90 and 94% in preterm babies receiving oxygen, with alarm limits set to 89 and 95% [90].

The increasing evidence regarding the toxic effects of oxygen has led to a revision of oxygen administration during resuscitation in the delivery room and not just during neonatal care to infants who have completed newborn transition but still require supplemental oxygen. In the last thirty years, the approach to neonatal resuscitation and stabilization in the delivery room has been modified with difficulty due to the presence of controversial opinions regarding the optimal amount of oxygen to be administrated. Even though the World Health Organization changed its guidelines in 1998, stating that room air is adequate for basic resuscitation [90], the International Liaison Committee on Resuscitation (ILCOR) and the American Heart Association (AHA)/American Academy of Pediatrics (AAP) stated that 100% oxygen is the first choice for reoxygenation of depressed newborn [91,92].

Since the 1990s five clinical studies randomized newborns in need of resuscitation to 21% e 100% oxygen demonstrating that resuscitation can be performed by using room air. These studies highlighted that the use of 100% oxygen increases the time to first breath by approximately 24 s in one study [93] and by 1 min in another [94], it also increases the time to establish normal regular breathing [95], the duration of resuscitation [96] and oxidative stress with long-term effects. In one of these studies, Vento et al. calculated the ratio between reduced and oxidized glutathione in erythrocytes and the urinary output of 8 OH-DH, a marker of oxidative stress of mitochondrial DNA. Authors found the presence of elevated oxidative stress not only immediately after birth but at least 4 weeks after birth in infants given 100% oxygen, otherwise oxidative stress normalizes within a few days in infants randomized with 21% oxygen [97]. Resuscitation with room air also reduces neonatal mortality, in fact, neonatal mortality rate is 13% and 8% respectively in the group 100% O_2_ and 21% O_2_.

Several recent different meta-analyses of randomized and quasi-randomized trials showed that short-term mortality is reduced when respiratory support of term and late preterm neonates is started in room air compared with 100% oxygen [98] and no difference is described in moderate to severe neurodevelopmental impairment at 1 to 3 years of age for newborns administered 21% oxygen,. Similarly other meta-analysis of randomized trials enrolling preterm newborns showed no difference in short-term and long-term mortality, neurodevelopmental outcome, ROP, BPD, NEC, or major cerebral hemorrhage when respiratory support was started with low oxygen concentrations [99,100].

To reduce the toxic effects associated with both to hypoxia and hyperoxia, furthermore in consideration of the data provided by the latest studies, the American Heart Association (AHA) updated guidelines for optimal management of oxygen during resuscitation. AHA stated that room air (21% oxygen) should be used to initiate resuscitation in term and late preterm neonates and that a lower oxygen concentration (21–30%) is recommended to initiate resuscitation of preterm neonates (<35 weeks’ gestation) [101].

When administering oxygen during stabilization at the delivery room it is necessary to remember that during the transition from intrauterine to extrauterine life blood oxygen levels rise gradually due to the presence of residual pulmonary shunts. In most healthy term and near-term neonates born during an uncomplicated delivery it took almost 5 min to reach an oxygen saturation >80% and almost 8 min to reach a median pre-ductal SpO2 of 90%. These data suggest that normal infants take some time to transition and that mimicking the normal rate of increase in blood oxygenation during resuscitation may reduce oxidative injury [102,103,104].

Accordingly, following the results of the two trials, STOP-ROP and NeOProM, the oxygenation monitoring in NICU, the SpO2 target range should be approximately 90–95% Accepting these results, most NICUs have used supplemental oxygen maintaining an SpO2 range of 89–95%, since the late 2010s [104,105]. Therefore, extreme excess supplementation of oxygen is not administered in modern NICUs. This means that ROS-mediated injuries due to Hyperoxia may be reduced in the recent era.

## 8. Conclusions

Supplemental oxygen may be necessary during resuscitation and beyond stabilization, especially in preterm newborns, in order to prevent the consequences of hypoxemia, however, oxygen must be used with caution to avoid the damage associated with hyperoxia. The increasing knowledge about the toxic effects of supplemental oxygen and neonatal susceptibility to oxidative stress has led to a significant review of guidelines and clinical practice.

According to numerous studies about the toxic effects of supplemental oxygen, clinical use of oxygen has been progressively reduced in favor of ventilation in the last twenty years. Despite recent clinical studies, oxygen can still be considered a double-edged sword since higher levels may increase survival, although with an increased risk of severe sequalae as ROP and BPD. On the other hand, lower oxygen levels are associated with an increased risk of death while oxygen-related morbidities are lower. More studies are required to optimize oxygen supplementation in order to maximize the beneficial effects while minimizing the harmful ones. 

## Figures and Tables

**Figure 1 children-10-00579-f001:**
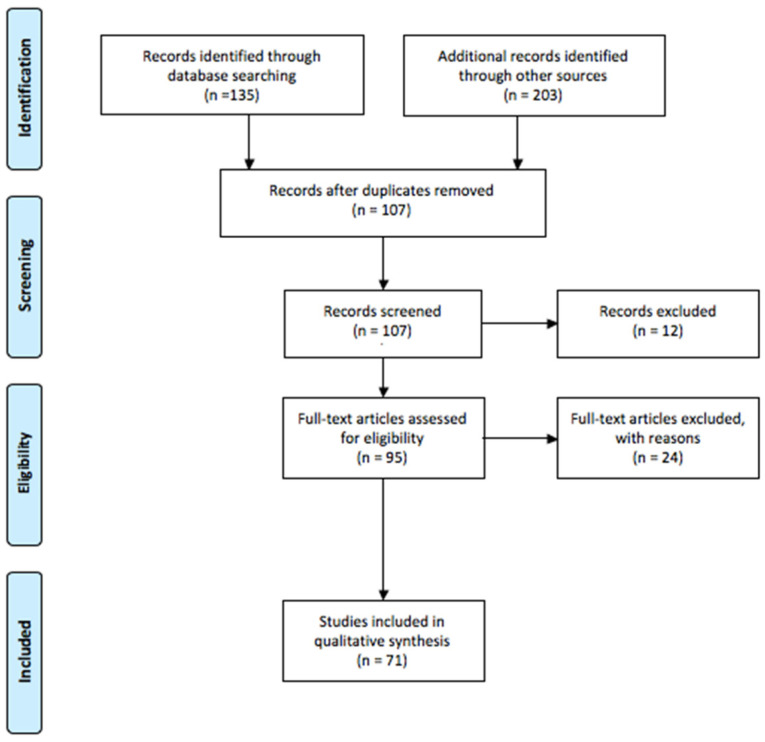
Flow chart of the literature research for two independent reviewers.

**Figure 2 children-10-00579-f002:**
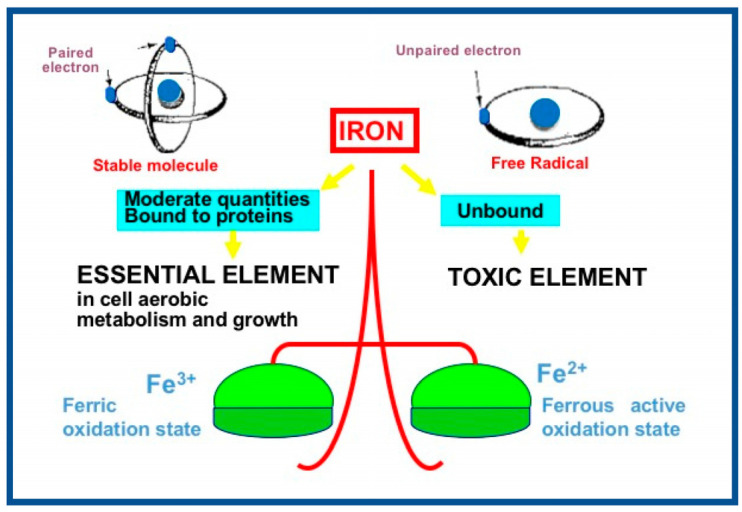
Schematic representation of iron redox state. Iron is a versatile and highly reactive element. Having two common valences, ferrous iron Fe^2+^ and ferric iron Fe^3+^, it has access to a wide range of redox (oxidation/reduction) potential.

**Table 1 children-10-00579-t001:** List of the included papers. Table legend: OS (oxidative stress); ROP (retinopathy of prematurity); VEGF (endothelial growth factor); TOS (Total oxidative status); TAS (total antioxidant status); MDA (malondialdehyde); PON1 (paraoxonase 1); PVL (periventricular leukomalacia); IPs (isoprostane); BPD (bronchopulmonary dysplasia); NOX (NADPH oxidase); ROS (reactive oxygen species); AEC (alveolar epithelial cell); SP (surfactant protein); NEC (necrotizing enterocolitis); NPBI (non-protein bound iron); AOPP (advanced oxidation protein products); TH (total hydroperoxides; PDA (patent ductus arteriosus).

Disease	Reference	Population	OS Biomarker	Results
ROP	Pierce 1996 [56]	Neonatal mice exposed to hyperoxia	VEGF	The expression of VEGF in the peripheral retina was down-regulated by hyperoxia in conjunction with the arrest of growth and the loss of some of the developing vasculature.
	Banjac 2018 [51]	Preterm newborns	TOS, TAS, MDA and PON1	TOS and MDA were significantly higher in infants with ROP as compared to infants without ROP
WMI	Coviello 2021 [54]	Preterm newborns	Cord blood and plasma IPs	Cord blood IPs were not correlated with white matter injury score, whereas higher plasma IPs and lower gestational age predicted higher white matter injury score
	Coviello 2022 [55]	Preterm newborns	IPs	Higher plasma IPs levels are associated with decreased functional brain activity.
BPD	Carnesecchi 2009 [57]	Wild-type and NOX1- and NOX2-deficient mice	ROS	ROS production was reduced in lung from NOX1-deficient mice.
	Hou 2015 [58]	Newborn rats	AEC I (aquaporin 5, T1α) and AEC II markers (SP-C, SP-B)	Authors found an increase in AEC II-to-AEC I transdifferentiation in a hyperoxia-induced, BPD-like model at both the tissue and cellular levels.
NEC	Baregamian 2011 [59]	Rat and human fetal intestinal epithelial cells	ROS	OS leads to increased intracellular ROS production by mitochondria and activation of mitochondrial apoptotic signaling pathways in both human fetal and rat intestinal epithelial cells.
	Perrone 2012 [60]	Preterm newborns	NPBI, AOPP and TH	AOPP, TH and NPBI cord blood levels were significantly higher in babies with NEC
PDA	Longini 2010 [61]	Preterm newborns (<33 weeks of gestational age)	urinary IPs	Ibuprofen therapy for PDA closure reduces the risk of OS, inducing a decrease of urinary IPs.
	Chen 2012 [62]	Newborn mouse	IPs	IPs levels are increased shortly after birth in response to increased oxygen tension and this may serve as a novel physiological signal to stimulate postnatal PDA closure. Authors found that IPs have both vasoconstrictive and vasodilatory effects.

## Data Availability

The data presented in this study are available on request from the corresponding author.

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
