# Peer review of "Oxygen for the Newborn: Friend or Foe?"

_children, 2023, doi:10.3390/children10030579_

Round 1

Reviewer 1 Report

This manuscript adds a detailed overview about oxygen supplementation in neonates. The review is very structured and comprehensively writen. 

Methods: 

There should be detailed information about the methods of the review:

1) A section on information on the search strategy is missing. Which search terms were used?

2) Has the review been registered in Prospero? 

Results:

1) A flow chart on excluded papers and finally included papers would be helpful. 

2) Please provide a table on included papers providing the most important information. 

Author Response

Reviewer 1
This manuscript adds a detailed overview about oxygen supplementation in neonates. The review is very structured and comprehensively written. 

Methods: 

There should be detailed information about the methods of the review:

1) A section on information on the search strategy is missing. Which search terms were used?

Re: We thank the reviewer for this observation. This was a narrative review, so there were no predetermined research question or specified search strategy, just a topic of interest. With this narrative review we intended to give a panoramic view of this topic, addressing every aspect of it and providing the newest knowledge of the topic.

The search terms were reactive oxygen species, oxidative stress, free radicals diseases of prematurity, newborn infants.

2) Has the review been registered in Prospero? 

Re: This was a narrative and educational review. It wasn’t registered in Prospero

Results

1) A flow chart on excluded papers and finally included papers would be helpful. 

Re: Flow chart of the literature research for two independent reviewers has been added (Figure 1)

2) Please provide a table on included papers providing the most important information. 

Re: A Table 1 on included papers providing the most important information has been provided

Author Response

Reviewer 2

Lines 43-190: It would be better to make a logistic order to understand readers. There

are some lacks in the connections between paragraphs. It mixes up details. Including

new information rather than already established know details of ROS and antioxidants

would be better.

Re: We thank the reviewer for this helpful remark. The text has been extensively reviewed following the suggestion.

Line 81: The production of the mechanism of ROS by mitochondria needs to be

clarified.

Re: Mechanisms of ROS production have been clearlier stated in text, through extensive revision

Line 103: Free iron is not always toxic in the biological system. Some conditions favour the harmful. It would be clearly stated.

Re: we thank the reviewer for this important suggestion. A paragraph containing these considerations has been provided. A Figure has been also provided (Figure 2).

Lines 189-190: It would be better to mention the prooxidant activity of vitamin E and the

antioxidant activity of vitamin C, which are more relevant to your review related to

oxidation.

Re: New sentences and relative references, deeping the information about prooxidant activity of vitamin E and the antioxidant activity of vitamin C, have been provided

Lines 125-126: More details on the molecular mechanism of ROS production are

needed by NOX families to understand readers.

Re: More details on the molecular mechanism of ROS production by NOX families have been provided

Lines 18-19: The meaning of 'antioxidant factors' is a bit confusing. Better to reword it.

Re: The sentence has been rephrased

Abstract- Line 25: it would be better to avoid the abbreviations in the keywords.

Re: The abbrevations have been deleted

Line 44: Reword as reactive oxygen "AND" nitrogen species.

Re: The word “and” has been added

Lines 53-57: The diagrammatic explanation would be easier to understand.

Re: The sentences have been clearly stated

Lines 98: Succinate dehydrogenase is a complex II. Please correct it.

Re: The mistake has been corrected

Line 119: What is NOX? Please write in complete form when it appears the first time.

Re: The complete form of NOX has been provided at the first appear in the text

Line 134: It would be better to revise the subtitle as "Functions of ROS" based on the

paragraph's content, which contains ROS functions. Line 146: The appearance of H2O2

is not connected to this para. Please revise it.

Re: The subtitle has been revised following the suggestion

Line 201: Provide the abbreviation as ROS.

Re: The abbreviation has been provided

Line 357: It would be better to add original research papers that dealt with oxygenation and the outcome related to oxidants and antioxidants status, if any

Re: We thank again the reviewer. Original research papers that dealt with oxygenation and outcome related to oxidant and antioxidant status have been considered. A table with the main relevant information has been provided

Round 2

Reviewer 2 Report

Thank you for the clarifications and extensive revision of the paper. 

Adding or revising the following minor typo and other issues would be better. 

Comments: 

Line 140: It would be better to add a detailed caption for figure 2 to connect the unpaired or paired oxygen to iron.

Line 173 and 286: Using the subscript for H2O2 and superscript for Fe2+ and Cu+ would be better.

Line 250 &254: Please revise "Vitamin C" to "vitamin C"

Line 296: IVH needs to be written in full form. 

Author Response

Reviewer Comments:

Reviewer 2

Thank you for the clarifications and extensive revision of the paper. 

Adding or revising the following minor typo and other issues would be better. 

Comments: 

Line 140: It would be better to add a detailed caption for figure 2 to connect the unpaired or paired oxygen to iron.

 Re: We thank the reviewer for this observation. A detailed caption for Figure 2 has been provided

Line 173 and 286: Using the subscript for H2O2 and superscript for Fe2+ and Cu+ would be better.

Re: The superscript and subscript have been checked and corrected through the paper

Line 250 &254: Please revise "Vitamin C" to "vitamin C"

Re: ‘vitamin C’ instead of Vitamin C has been used

Line 296: IVH needs to be written in full form. 

Re: The term IVH has been written in extensive form: intraventicular hemorrhage